# Updates on Therapeutic Strategies in the Treatment of Relapsed/Refractory Multiple Myeloma

**DOI:** 10.3390/cancers16172931

**Published:** 2024-08-23

**Authors:** Deevyashali S. Parekh, Yun Kyoung Ryu Tiger, Kevin Tony Jamouss, Justin Hassani, Maroun Bou Zerdan, Shahzad Raza

**Affiliations:** 1Department of Medicine, SUNY Upstate Medical University, Syracuse, NY 13210, USA; parekhd@upstate.edu (D.S.P.); bouzerdm@upstate.edu (M.B.Z.); 2Rutgers Cancer Institute, New Brunswick, NJ 08901, USA; yt467@cinj.rutgers.edu; 3University of Massachusetts Chan Medical School, Baystate Campus, Springfield, MA 01107, USA; kevin.jamouss@baystatehealth.org (K.T.J.); justin.hassani@baystatehealth.org (J.H.); 4Department of Hematology and Oncology, Cleveland Clinic, Cleveland, OH 44195, USA

**Keywords:** multiple myeloma, relapsed refractory, celmods, proteasome inhibitors, bispecific antibodies, CAR-T, monoclonal antibodies

## Abstract

**Simple Summary:**

Multiple myeloma is a heterogenous hematological condition with a worldwide incidence of between 0.5–5 per 100,000. The standard front-line treatment algorithm suggests induction chemotherapy, early stem cell transplant in those eligible, followed by maintenance chemotherapy. Unfortunately, most people subsequently relapse and require salvage treatment. A wide array of options are available in the management of relapse/refractory multiple myeloma. Our review aims to describe the new, approved treatment modalities in this setting while describing data on the efficacy, tolerability, and nuances of each available option.

**Abstract:**

Multiple myeloma is a heterogeneous condition characterized by the proliferation of monoclonal B-cells, for which there is currently no curative treatment available. Relapses are, unfortunately, common after first-line treatment. While the prognosis for relapsed refractory multiple myeloma is generally poor, advances in the treatment of relapsed or refractory multiple myeloma offer hope. However, the expansion of effective options in targeted treatment offers renewed optimism and hope that patients who fail on older therapies may respond to newer modalities, which are often used in combination. We review currently approved and novel investigational agents classified by mechanisms of action, efficacy, approved setting, and adverse events. We delve into future directions of treatment for relapsed/refractory multiple myeloma, reviewing novel agents and therapeutic targets for the future.

## 1. Introduction

Multiple myeloma (MM) is a B-cell malignancy characterized by the proliferation of monoclonal plasma cells in bone marrow and extramedullary sites. The International Myeloma Working Group defines relapsed/refractory multiple myeloma (RRMM) as progressive disease, inadequate response while on treatment, disease progression within 60 days of completion of most recent treatment in a patient who had achieved remission, absence of even a minimal response, or primary refractory disease.

MM is a heterogeneous condition for which there is currently no established cure. Relapses are common after first-line treatment, requiring new lines of treatment, but the probability of remission decreases with each subsequent line of therapy. Advances in treatment over recent years have improved the prognosis for some patients, as those who fail on older treatments with immunomodulatory drugs and proteasome inhibitors may respond to treatment with monoclonal antibodies, bispecific antibodies, chimeric antigen receptor T cells, and other newer agents. Newer treatment modalities offer renewed optimism for improved prognosis.

In this review, we will provide an overview of approved and investigational therapies for RRMM. 

## 2. Treatment of Relapsed Refractory MM

Treatment strategies for RRMM can be individually tailored for optimization. Different combinations of drugs available in this setting can be used at full or adjusted doses with consideration of prior resistance to a given drug class, previous or current toxicity from therapy, age, and comorbidities.

Targeted therapies take advantage of certain features of MM such as specific antigens that are highly expressed on myeloma cells and targeted by antibodies and CAR-T cells. Other approaches address mechanisms of myeloma cell resistance to therapy such as proteasome inhibitors to prevent the breakdown of pro-apoptotic proteins and thus preserve that immune defense. Two-drug and three-drug combinations of targeted therapies are based on clinical experience in RRMM populations and on individual patient variables such as disease staging and tolerability to adverse effects with different agents.

The following sections review approved and investigational treatments for RRMM within each category, defined by mechanisms of action. Table 1 summarizes currently approved treatments. Table 2 summarizes pivotal trials for commonly used regimens in RRMM.

### 2.1. Immunomodulatory Drugs (IMiDs)

The main mechanism of IMiDs in the treatment of MM is downregulation of Ikaros and Aiolos, proteins with diverse effects in leukocytes, including malignant cells and immune cells. In contrast to the inhibited tumor-suppressive effects of these proteins in acute leukemia, they are overexpressed in MM, promoting malignant cell growth, and impairing immune function [31]. IMiD binding to the protein cereblon marks the Ikaros and Aiolos transcription factors for degradation by the proteasome. The clinical result is direct myeloma cell destruction and the potentiation of immune responses to the malignancy.

Lenalidomide, analogous to thalidomide but with a different safety profile, has emerged as a mainstay in the initial treatment of MM. It is often used in combination with dexamethasone and the proteasome inhibitor bortezomib as an induction treatment before allogeneic HSCT, and in combination with dexamethasone and the anti-CD38 monoclonal antibody daratumumab as the first-line treatment for HSCT-ineligible MM patients. However, its role is more limited in RRMM patients, whose prior treatments have often included an IMiD. Early reports showed that lenalidomide in combination with dexamethasone is more effective than dexamethasone alone in the treatment of RRMM [32,33]; however, subsequent research typically utilizes lenalidomide–dexamethasone as a control group to assess outcomes with the addition of a third agent such as daratumumab [34], the proteasome inhibitors carfilzomib [1] or ixazomib [2], or the anti-SLAMF7 antibody elotuzumab [15].

Pomalidomide is approved for use in RRMM patients who have received at least two previous therapies including lenalidomide and a proteasome inhibitor. As part of triplet regimens, pomalidomide has shown promising efficacy and tolerability [16,35,36]. An additional advantage of this agent is hepatic clearance, which makes it especially suitable in RRMM patients with renal dysfunction.

### 2.2. CELMoDs

A common cause of treatment failure in MM is resistance to IMiDs. Cereblon E3 ligase modulators (CELMoDs) are investigational derivatives of IMiDs that exhibit greater binding affinity to cereblon and more potent antitumor activity. This modality has improved the prognosis of MM patients who have become resistant to the original ImiDs [37,38]. 

Mezigdomide (CC-92480), in combination with dexamethasone, was assessed in a Phase I/II study in patients with triple-class refractory RRMM. The treatment results included an overall response rate (ORR) of 41%; a median duration of response (DOR) of 7.6 months; and median progression-free survival (PFS) of 4.4 months. The most common adverse effects were neutropenia and infection [39].

Iberdomide (CC-220) in combination with dexamethasone is being assessed in an ongoing multicenter Phase I/II study in RRMM patients who had received at least two previous lines of therapy, including an IMiD and a proteasome inhibitor (NCT02773030). Interim data show that the regimen produces meaningful clinical activity (ORR 32% in the dose-escalation cohort, 26% in the dose-expansion cohort) and is generally safe; the most commonly reported adverse events are neutropenic infection, anemia, and thrombocytopenia [38].

### 2.3. Proteasome Inhibitors

Proteasomes are large protein complexes that degrade damaged or unneeded proteins through the action of protease enzymes that break peptide bonds. Cancer cells evade programmed cell death by breaking down pro-apoptotic proteins. In cancer treatment, proteasome inhibitors (PIs) interact with specific proteasome sites to prevent the breakdown of pro-apoptotic proteins. The result is cell cycle arrest and the induction of apoptosis in malignant cells. Common adverse effects of PIs include gastrointestinal disturbances, headache, and fatigue.

Bortezomib was approved for use in MM in the US in 2003 and in Europe in 2004. It is administered intravenously. In the Phase III VISTA trial, treatment-naïve MM patients were given melphalan–prednisone with or without the addition of bortezomib. The patients receiving bortezomib showed significantly longer time to progression (TTP) and significantly higher ORR [40]. In the Phase III APEX study comparing bortezomib to high-dose dexamethasone in patients with RRMM, bortezomib was associated with significantly longer TTP, higher response rates, and improved survival. A subgroup analysis showed a similar advantage with bortezomib in elderly and high-risk patients [41].

Carfilzomib (approved in 2012) has been studied most often as part of a triplet combination. In a Phase II trial, outcomes in RRMM patients receiving weekly treatment with carfilzomib, cyclophosphamide, and dexamethasone included ORR 85%, median overall survival (OS) 27 months, and a PFS of 17 months, with poorer outcomes in patients with high-risk cytogenetics [42]. A retrospective analysis of outcome data from 197 RRMM patients treated with carfilzomib, lenalidomide, and dexamethasone showed ORR 88%, a median PFS of 19.8 months, and a 1-year OS rate of 80.6%, with poorer outcomes in patients with high-risk cytogenetics. Neutropenia and infection were the most common serious adverse effects [43]. A summary of systematic reviews of clinical trials concluded that carfilzomib significantly improves PFS, OS, and ORR in patients with MMRR but incurs the risk of cardiotoxicity, nephrotoxicity, and infection [44].

Ixazomib (vs placebo), in combination with lenalidomide and dexamethasone, was assessed in the Phase III TOURMALINE-MM1 trial in RRMM patients. The use of ixazomib resulted in significantly longer PFS (20.6 vs. 14.7 months) with limited additional toxicity [2]; however, long-term follow-up analysis of median OS showed an advantage with ixazomib only in certain predefined subgroups, as follows: treatment-refractory, age >65, stage III disease, >1 prior therapy, and high-risk cytogenetics [3]. Ixazomib was approved in the US in 2015 and in Europe in 2016.

### 2.4. XPO1 Inhibitor

Exportin-1 (XPO1; also called chromosome region maintenance 1) is a nuclear export protein that regulates transport of proteins and RNA. XPO1 is often overexpressed in MM, which enables cancer cells to inactivate tumor suppression proteins (TSPs) by exporting them to the cytoplasm. Forced nuclear retention of TSPs and mRNA for oncogenic proteins through XPO1 inhibition represents a unique therapeutic strategy, by keeping TSPs active while impeding formation of oncoproteins [45]. In vitro studies in myeloma cells confirm that XPO1 inhibition results in increased apoptosis and reduced synthesis of oncoproteins.

The oral XPO1 inhibitor selinexor in combination with dexamethasone was evaluated in the Phase II STORM trial in MM patients who had previously received bortezomib, carfilzomib, lenalidomide, pomalidomide, daratumumab, and an alkylating agent, and were triple-refractory to proteasome inhibitor, immunomodulatory agent, and daratumumab. ORR was 26%; median DOR was 4.4 months, median PFS was 3.7 months, and median OS was 8.6 months [46]. The Phase III BOSTON trial compared once-weekly selinexor-bortezomib-dexamethasone vs. twice-weekly bortezomib-dexamethasone in patients with previously treated MM. The selinexor regimen showed significant therapeutic advantage in median PFS, 13.9 vs. 9.5 months [27]. Common adverse reactions with selinexor include gastrointestinal disturbances, fatigue, peripheral neuropathy, and upper respiratory infection; hematologic abnormalities include anemia and decrease in thrombocytes, lymphocytes, and neutrophils.

Selinexor is FDA-approved for use in combination with bortezomib and dexamethasone in treating MM in adults who have received at least one prior therapy; and in combination with dexamethasone in adults with RRMM who have received at least four prior therapies and whose disease is refractory to at least two proteasome inhibitors, at least two immunomodulatory agents, and an anti-CD38 monoclonal antibody.

### 2.5. BCL2 Inhibitor

The B-cell lymphoma-2 (BCL-2) protein, often elevated in MM, acts as an oncogene by binding pro-apoptotic proteins in myeloma cells. BCL-2 inhibition therefore represents a novel therapeutic strategy, which may be especially important in the subset of MM patients (variably estimated at 15%–24%) whose plasma cells are heavily dependent on BCL-2 for survival due to a t(11;14) gene translocation.

Venetoclax is a selective BCL-2 inhibitor (approved for chronic lymphocytic leukemia and small lymphocytic lymphoma, investigational in MM). By binding to BCL-2, venetoclax displaces pro-apoptotic proteins, which then induce apoptosis in the malignant cell. In a Phase I trial in heavily pretreated RRMM patients, venetoclax monotherapy yielded a 21% ORR, but response was markedly higher in the subset of patients who were positive for t(11;14) [47]. Similarly, in a Phase II study in t(11;14)-positive RRMM patients, venetoclax plus dexamethasone yielded a 48% ORR. The Phase III BELLINI study assessed venetoclax-vs-placebo in combination with bortezomib-dexamethasone in RRMM patients. Median PFS was significantly longer with venetoclax than with placebo (22.4 vs. 11.5 months) but mortality was higher due to an increased incidence of infection [29]; on subgroup analysis, this risk did not show correlation with the t(11:14) translocation.

Lisaftoclax (APG-2575) is an investigational BCL-2 inhibitor previously studied in chronic lymphocytic leukemia. In a Phase I study in patients with RRMM or light-chain amyloidosis (NCT04942067), lisaftoclax was given in combination with pomalidomide-dexamethasone (Arm A) or lenalidomide-dexamethasone-daratumumab (Arm B). Interim data from the first 30 patients (25 RRMM: 22 in Arm A, 3 in Arm B) showed generally good tolerability. Partial response or better was reported in 14 of 21 evaluable patients in Arm A and in 2 of 3 in Arm B [48].

### 2.6. Monoclonal Antibodies

This type of immunotherapy utilizes antibodies derived from a single clone for precise targeting of specific antigens on malignant cells, activating and enhancing the patient’s immune system. Binding to a specific antigen on targeted cells results in antitumor action through induced apoptosis or direct cytotoxic activity, antibody-dependent cellular phagocytosis, or complement-dependent cytotoxicity. With intravenous administration, typical side effects include infusion reactions (especially at initial infusion), generalized symptoms like fever and chills, and gastrointestinal disturbances; adverse effects can also occur if the targeted antigen is found on normal cells that affect specific functions (for example, antibody binding to vascular endothelial growth factor could cause problems involving bleeding and blood pressure regulation).

#### 2.6.1. Monoclonal Antibodies Targeting CD38 on Myeloma Cells

CD38 antigen is highly expressed on myeloma cells and, to a lesser extent, on regulatory T cells, regulatory B cells, and myeloid-derived suppressor cells. Monoclonal antibodies that target CD38 bring about direct antitumor activity. Two IgG anti-CD38 monoclonal antibodies are currently approved for use in treating MM.

Daratumumab monotherapy was tested in the GEN501 and SIRIUS trials in 148 RRMM patients. In a pooled analysis of outcomes from these trials, ORR was 30.4% and median OS was 20.5 months [49]. Two Phase III trials in RRMM patients showed benefits with daratumumab in combined therapy: In the POLLUX trial, OS was significantly longer with addition of daratumumab to lenalidomide-dexamethasone [9]; and in the CASTOR trial, PFS and OS were significantly longer with addition of daratumumab to bortezomib-dexamethasone [8]. Daratumumab received FDA approval in 2015.

Isatuximab, is similar to daratumumab in its antitumor effects but also directly induces apoptosis in myeloma cells. Significantly improved PFS in RRMM patients was demonstrated in the Phase III ICARIA-MM trial of pomalidomide-dexamethasone plus isatuximab [12] and the Phase III IKEMA trial of carfilzomib-dexamethasone plus isatuximab [13]. Isatuximab was approved in 2020 for use in combination with pomalidomide-dexamethasone or carfilzomib-dexamethasone in RRMM patients.

Felzartamab (MOR202), an investigational anti-CD38 monoclonal antibody, demonstrated acceptable safety and tolerability alone and in combinations with dexamethasone, dexamethasone-pomalidomide, or dexamethasone-lenalidomide in a dose-finding trial (NCT01421186) in RRMM patients [50].

Mezagitamab (TAK-079) was assessed as monotherapy in a Phase I dose-escalation study (NCT03439280) in patients with RRMM. Interim data on patients receiving at least four cycles of therapy showed an ORR of 43%, with an ORR of 46% in patients with no previous anti-CD38 therapy [51]. 

#### 2.6.2. Monoclonal Antibodies Targeting Other Myeloma Cell Antigens

Elotuzumab targets the signaling lymphocyte activation molecular family 7 (SLAMF7) receptor, which is expressed mainly on malignant and normal plasma cells. Although ineffective as monotherapy, elotuzumab was approved in 2018 for use in combination with pomalidomide–dexamethasone or lenalidomide–dexamethasone in patients with RRMM. In the Phase II ELOQUENT-3 trial, RRMM patients previously treated with lenalidomide and a proteasome inhibitor showed significant improvements in OS and PFS with the addition of elotuzumab to pomalidomide–dexamethasone [16]. A meta-analysis of five randomized controlled trials showed that the addition of elotuzumab to pomalidomide–dexamethasone or lenalidomide–dexamethasone improved OS and PFS in patients with RRMM but did not reduce the risk of disease progression in patients with newly diagnosed MM; the incidence of serious adverse events was higher with the elotuzumab combinations [52].

Pembrolizumab, an anti-PD-1/PD-L1 antibody approved for a variety of solid tumors, was investigated for use in MM, based on myeloma cells’ dependence on PD-1/PD-L1 interaction to evade immune defenses. However, in the KEYNOTE-183 trial, the risks outweighed the benefits with the addition of pembrolizumab to pomalidomide–dexamethasone in patients with RRMM [53]. Similarly, in the KEYNOTE-185 trial, the risk–benefit ratio was unfavorable with the addition of pembrolizumab to lenalidomide–dexamethasone in HSCT-ineligible patients with newly diagnosed MM [54].

### 2.7. Bispecific Antibodies

Bispecific antibodies (also called bispecific T cell engagers) bind simultaneously to a target antigen on tumor cells and to the CD3 receptor on cytotoxic T cells. As a result, the T cell is activated to kill the linked tumor cell. However, the effectiveness of this modality may be limited by immunosuppressive effects of the tumor microenvironment (potentially offset by co-therapy with drugs that limit those effects) and by serious adverse events, including cytokine release syndrome (CRS: the rapid release of cytokines into the blood, causing diverse and sometimes severe symptoms) and infections [55]. 

#### 2.7.1. BCMA-Directed Bispecific Antibodies

A main target for bispecific antibodies is B-cell maturation antigen (BCMA), which is overexpressed in MM. Teclistamab, approved in 2022, was assessed in the Phase I/II MajesTEC trial in heavily pretreated RRMM patients. The ORR was 65% and the median PFS was 11.3 months. CRS occurred in 72% of patients but most cases were Grade 1 or 2. Neutropenia, anemia, thrombocytopenia, and infection were also commonly reported [17]. In a review of outcome data from MajesTEC vs. four clinical trials in which RRMM patients received physicians’ choice of treatment after discontinuation of daratumumab, teclistamab showed significant advantage in ORR, and very good or better partial response, OS, and PFS [53]. LimiTEC is a non-inferiority study in which RRMM patients who had very good or better partial response after 6–9 months of teclistamab will be taken off treatment and monitored for up to 24 months to assess the risk of disease progression and adverse effects in comparison with ongoing treatment [49].

Elranatamab was approved in 2023 for adult RRMM patients who have received at least four prior lines of therapy including a PI, an IMiD, and an anti-CD38 monoclonal antibody. In the MagnetisMM-1 trial in heavily pretreated RRMM patients, the median PFS was 11.8 months and the median OS was 21.2 months; at median 12-month follow-up, the ORR was 63.6% and 38.2% of patients achieved complete response (CR). Adverse events included cytopenias and CRS, but no dose-limiting toxicities were reported [33]. In the Phase II MagnetisMM-3 trial, patients who responded well to weekly treatment with elranatamab were switched to biweekly treatment to improve tolerability. The ORR was 61.0% and the CR rate was 35.0%. At 15 months, the DOR rate was 71.5%, the PFS rate was 50.9%, and the OS rate was 56.7% (medians were not reached). The most commonly reported adverse events were infections, CRS, anemia, and neutropenia [19,20].

Several BCMA-directed bispecific antibodies are currently at the investigative stage for use in MM. AMG 420 links T cells to tumor cells by attachment to two target antigens: BCMA and CD19. In a dose-escalation study in RRMM patients, the ORR was 31% across dosages and was 70% at the maximum tolerated dose. Almost half the patients experienced serious adverse events, including infection and polyneuropathy [56].

AMG701, offering an extended half-life, produced effective cytotoxicity in an in vivo model of MM [57] and showed enhanced potency when co-administered with IMiDs in preclinical models [58]. In a dose-escalation study in heavily pretreated RRMM patients, the ORR was 36% across dosages and 83% at the most favorable dosage. The median DOR was 3.8 months. CRS (mostly grade 1 or 2) and cytopenias were commonly reported [59]. 

REGN5458 (linvoseltamab) was assessed at two doses in the LINKER-MM1 Phase II study, achieving numerically higher ORR at the higher dosage (64% vs. 50%) and incurring comparably high incidences of CRS (37% vs. 53%, mostly grade 1 or 2) [60]. In other research, this agent achieved efficacy comparable to anti-BCMA CAR-T cells, with a more rapid onset of effect [31].

REGN5459 is being tested in RRMM patients in a Phase I/II trial (NCT04083534). Interim data show an ORR of 65.1% and 90.5% at the high dosage. Median DOR was not reached but the probability of effect lasting at least 12 months was 78.1%. More than half the patients developed CRS, mostly grade 1 [32].

CC-93269 (BMS-986349; alnuctamab) was administered by two different routes in Phase I research in RRMM patients (NCT03486067). Interim data with intravenous (IV) administration show ORR 39%, median DOR 33.6 months, and incidence of CRS 76%; subcutaneous (SC) administration was more tolerable, allowing more optimal dosing and greater response (ORR 53% across dosages, 65% at target dosage, incidence of CRS 56%) [61]. Follow-up data confirmed the advantage of SC administration: ORR 54% across dosages, 69% at target dosage, median PFS 10.1 months, and incidence of CRS 56% [62].

ABBV-383, administered IV, was assessed in a Phase I dose escalation/expansion trial in RRMM patients. ORR was 57% across dosages, 68% at the target dosage range, and incidence of CRS was 57% [63]. In an update report on 3 dose levels (20, 40, and 60 mg) assessed in the expansion phase of the trial, treatment-emergent adverse events were frequent but manageable at all 3 dosages but treatment effectiveness (measured by estimates of PFS and DOR) was lower at 20 mg than at 40 and 60 mg [64].

F182112, administered IV, is being assessed in a Phase I dose-escalation study in RRMM patients (NCT04984434). In an interim report on 22 treated patients at median 3-month follow-up, low-grade CRS and cytopenias were the most commonly reported adverse events. Among 20 patients assessed for response to treatment at 3, 10, 20, and 30 μg/kg, ORR was 45% across dosages and 78% at the 10- and 20-μg/kg dosages [65].

WVT078 is being assessed in combination with WHG626, a γ-secretase inhibitor in a Phase I study in RRMM patients (NCT04123418). In theory, WHG626 may enhance anti-BCMA effectiveness at lower dosages by preventing cleavage of the target antigen by γ-secretase. In an interim report on the first 23 patients, 18 discontinued treatment (mostly for disease progression) and 5 experienced dose-limiting toxicities. ORR was 39.1% across dosages and 57.1% at the two highest dosages. Pharmacodynamic biomarkers confirmed the theorized mechanism of this combined therapy [66].

#### 2.7.2. Non-BCMA-Directed Bispecific Antibodies

Bispecific antibodies directed at myeloma cell antigens other than BCMA have shown promise in heavily pretreated RRMM patients, including those with previous exposure to BCMA-directed therapies [67].

Talquetamab binds to G-protein-coupled receptor family C, group 5, member D (GPRC5D, an orphan receptor found in normal plasma cells and myeloma cell lines) and to CD3 on T cells. In the Phase I MonumenTAL-1 study in heavily pretreated RRMM patients, talquetamab achieved ORR 70% with IV administration and 64% with SC administration, median DOR 10.2 and 7.8 months, CRS incidence 77% and 80% [68]. Follow-up data showed continuing effectiveness (ORR 73%, median DOR > 1 year) [21]. Among patients who went on to a subsequent line of therapy after talquetamab, the best outcomes were seen with CAR-T (ORR 66.7%, CR rate 25%) [69]. In the Phase II MonumenTAL-2 study, talquetamab in combination with pomalidomide was effective and well tolerated in RRMM patients [70]. Talquetamab is currently the only FDA-approved non-BCMA-directed bispecific antibody.

Forimtamag also binds to GPRC5D and CD3. In a dose-escalation trial in RRMM patients previously treated with IMiDs and PIs, treatment with forimtamag achieved 66.7% ORR across dosages, with most responses rated as very good or better; response was more modest in patients previously treated with BCMA-directed therapy. Median DOR was 12.2 months [71].

Cevostamab (BFCR4350A) binds to the tumor-associated antigen Fc receptor-like protein 5 (FcRH5) on myeloma cells and to CD3 on T cells. Interim data from a Phase I dose-escalation trial in RRMM patients (NCT03275103) show adverse events in almost all patients, with low-grade CRS occurring most often in cycle 1. Among efficacy-evaluable patients, ORR was 51.7% across dosages [72].

AMG 424 targets myeloma cells expressing CD38 antigen for linkage to CD3 on T cells. Preclinical research has shown potent activity and attenuated cytokine release [73]. However, a Phase I trial in RRMM patients (NCT03445663) was terminated by the manufacturer, with no results posted.

### 2.8. CAR-T Cells

Chimeric antigen receptor (CAR) T cells are obtained from the patient (autologous) or from donors (allogeneic, for “off the shelf” use). The cells are modified in the laboratory to express a surface receptor that recognizes and binds to a specific cell-surface antigen present on tumor cells. The most common antigenic target for CAR-T cells is BCMA. The modified T cells are reproduced in great numbers and then infused into the patient, where they bind to tumor cells expressing the target antigen, resulting in lysis of the tumor cell. A meta-analysis of 22 clinical trials of CAR-T therapy in RRMM reported a pooled ORR of 82%, a CR rate of 47%, a median PFS of 14 months, a median OS of 24 months, and low rates of CRS and neurotoxicity [74]. 

#### 2.8.1. BCMA-Directed CAR-T Cells

Ide-cel (idecabtagene vicleucel) was the first CAR-T therapy approved for the treatment of MM. In the Phase II karMMa trial in RRMM patients (minimum three lines of prior therapy), the ORR was 73%, the CR rate was 33%, and the median PFS was 8.8 months. The majority of patients experienced CRS and/or cytopenias [22]. In the Phase III karMMa-3 trial, RRMM patients treated with Ide-cel showed significantly better responses, as follows, than those treated with conventional therapy (one of five standard regimens): a median PFS of 13.3 vs. 4.4 months; an ORR of 71% vs. 42%; and a CR rate of 39% vs. 5%. The incidence of adverse events was 93% (most often CRS) vs. 75% [23]. Findings from a Phase I trial suggest that the duration of response with Ide-cel may be longer with the addition of a PI3K inhibitor (excessive activity in the phosphoinositide 3-kinase signaling pathway assists in the survival and reproduction of malignant cells) [75].

Cilta-cel (ciltacabtagene autoleucel), approved for use in RRMM in 2022, was assessed in the Phase I/II CARTITUDE-1 trial in patients with RRMM (at least 3 lines of prior treatment). Cilta-cell was given as a single IV dose 5–7 days after beginning of a lymphodepletion regimen of cyclophosphamide and fludarabine. At median 12.4-month follow-up, ORR was 97% and CR rate was 67%. Medians for PFS and OS were not reached but rates of PFS and OS at 12 months were 77% and 89%, respectively. Most patients developed CRS and neutropenia, and six deaths (among 97 treated patients) were attributed to treatment [24]. Follow-up at median 27.7 months showed durable responses: ORR 97.9%, CR rate 82.5%, PFS rate 54.9%, OS rate 70.4%, with poorer response rates among patients with high-risk cytogenetics, stage III disease, high tumor burden, or plasmacytomas. No new safety concerns were noted [76].

Among investigational BCMA-directed CAR-T therapies, Orva-cel (orvacabtagene autoleucel) showed favorable outcome data in RRMM patients in the Phase I/II EVOLVE study [77], but the manufacturer discontinued development of this product in 2021.

CT053 was assessed in a Phase I study in 24 RRMM patients in China, who received one cycle of CT053 following lymphodepletion with cyclophosphamide and fludarabine. ORR was 87.5%, CR rate was 79.2%. Among patients who progressed, median PFS was 281 days; disease progression was associated with ECOG > 1, grade III disease, concomitant extramedullary disease, and lower hemoglobin at baseline. Cytopenias and CRS were commonly reported [78]. In a follow-up report, 15 of the original 24 patients dropped out before 2 years (13 due to disease progression). ORR and CR rate were unchanged, median PFS was 18.8 months, and median DOR was 21.8 months. Disease progression was associated with ECOG > 1, grade III disease, and high-risk cytogenetics. Again, cytopenias and low-grade CRS were the most commonly reported adverse events [79].

ALLO-715 (an allogeneic CAR T cell designed to minimize immune rejection) is being assessed in the Phase I UNIVERSAL trial in RRMM patients (NCT04093596). In an interim report, 43 patients received ALLO-715 infused at selected doses following lymphodepletion with cyclophosphamide, fludarabine, and the anti-CD52 antibody ALLO-647. Incidence of CRS was 55.8% (one grade ≥ 3); of neurotoxicity, 14% (no grade ≥ 3); of infection, 54% (10 grade ≥ 3). ORR was 56% across dosages, and 71% at the most favorable dosage. Median DOR was 8.3 months) [80].

P-BCMA-ALLO-1 is an allogeneic CAR-T cell with a high percentage of T memory stem cells (Tmsc), which are capable of self-renewal for prolonged effect. It is engineered to prevent graft-vs-host responses. A Phase I study of dose-escalation (following lymphodepletion) is assessing safety and maximum tolerated dosage in RRMM patients (NCT04960579). Interim data on the first 24 patients show no dose-limiting toxicities and a low incidence of CRS and other adverse events, most of which were considered unrelated to treatment [81].

P-BCMA-101, an autologous CAR-T cell also engineered with a high percentage of Tmsc, was assessed in the Phase I/II PRIME study in RRMM patients (NCT03288493). In Phase I, patients received P-BCMA-101 (after a standard lymphodepletion regimen, and at a median dose determined by initial dose escalation) as a single treatment or biweekly infusion, alone or in combination with rituximab or lenalidomide to minimize autoimmune reaction to therapy. Toxicity was low and it appeared that lower doses might be more effective [82]. However, in 2024, the Phase II study was terminated by the manufacturer, with no results posted. 

#### 2.8.2. GPRC5D-Directed CAR-T Cells

BMS-986393 (CC-95266), an autologous CAR-T cell, was assessed in a Phase I dose-escalation trial in RRMM patients (NCT04674813). Following lymphodepletion, patients received a single infusion of BMS-986393 at various dosages to determine the maximum tolerated dose and recommended dosage for Phase II research. Interim data on the first 70 patients showed a 91% incidence of grade ≥3 adverse events including neutropenia; the CRS incidence was 84%, mostly grade <3. The ORR was 86% and the CR rate was 38% in all efficacy-evaluable patients, and 85% and 46% in patients who were refractory to previous BCMA-directed therapy [83]. 

MCARH109, having shown promising preclinical antitumor efficacy, was assessed in a Phase I dose-escalation trial in heavily pretreated RRMM patients (NCT04555551). At the highest dosage, grade ≥3 CRS and cerebellar were noted. Across dosages, the ORR was 71%, and responders included patients who had previously received BCMA-directed therapy [84]. A currently ongoing Phase I trial will assess the effect of adding MCARH109 to MCARH125, a BCMA-directed CAR-T cell therapy, in RRMM patients (NCT05431608).

## 3. Future Directions

As with any complex disease for which there is no definitive treatment, improved outcomes in RRMM may come about through the individualized selection of therapy that reflects the patient’s specific clinical characteristics. Analyses of outcomes in defined subpopulations from large-scale clinical trials can assist in planning therapy that takes into consideration the patient’s prior treatment, physical and laboratory findings, cytogenetics, disease staging, and demographic profile.

Another approach is the optimal use of available measures for the prevention and management of MM complications (e.g., bisphosphonates to reduce the accelerated rate of bone turnover) and treatment-induced toxicity (e.g., hematopoietic stimulants to minimize cytopenias).

Finally, the development and testing of novel therapeutic options could lead to effective MM treatment in patients who no longer respond to existing modalities. Several innovative strategies are currently being explored; examples are summarized below.

Newer therapies for RRMM have different mechanisms of action compared to previously approved therapies and hence have distinct toxicities associated with them that are also different. 

Cytokine release syndrome (CRS) is associated with all the mentioned CAR-T cell therapies and bispecific antibodies (Teclistamab, Talquetamab, Elranatamab) [17,20,21,22,76,80,85]. It is a strong systemic immune response characterized by increased levels of interleukin-6 (IL-6) and interferon gamma. It is graded based on the severity of hypoxia, fever and/or hypertension. The management of mild CRS involves supportive care. Systemic steroids and tocilizumab have been used in the management of severe or life-threatening CRS.

Immune effector cell-associated neurotoxicity syndrome (ICANS) is another systemic immune-response-based toxicity seen with CAR-T cell therapy and bispecific antibodies [17,22,80,85]. It occurs less frequently than CRS, has a widely variable presentation including, but not limited to, confusion, aphasia, agitation, and has been treated with dexamethasone.

Cilta-cel and Ide-cel have been associated with a parkinsonism-like movement disorder [86] and with hemophagocytic lymphohistiocytosis macrophage activation syndrome (HLH-MAS) [22,24].

Talquetamab has been associated with skin and nail changes and oral toxicities like loss or change in taste [87]. 

### 3.1. Conjoining Antibodies with Antitumor Toxins

One factor of treatment failure in RRMM may be dependance on the patient’s immune system, which may be compromised. An alternative strategy for bypassing such a dependence is immune conjugates—a monoclonal antibody linked to a cellular toxin that induces direct lysis of the targeted tumor cell. Two anti-CD38 antibodies engineered for linkage to anti-tumor toxins are TAK-169, which has shown promising results in preclinical research [88], and MT-0169, for which a Phase I dose-escalation study (NCT04017130) suggested possible utility in extramedullary MM.

Belantamab mafotodin, a BCMA-directed antibody conjugated to a toxin that kills myeloma cells through several mechanisms, had been approved as a first-in-class therapy that showed promising outcomes in combination with lenalidomide and dexamethasone in a small study of RRMM patients [89]. However, it was subsequently withdrawn from the US and Europe because outcomes from the Phase III research were considered unsatisfactory. Recently published data by the DREAMM-8 investigators [30] demonstrated an improved, durable, and deep PFS with Belantamab mafotodin, Pomalidomide and Dexamethasone (BPD) group compared to bortezomib, pomalidomide, and dexamethasone (PVd), especially amongst lenalidomide-exposed patients. It achieved this with a tradeoff of increased adverse events, including a higher rate of Grade 3 or worse adverse effects, with ocular events being more common; these were managed with Belantamab mafodotin dose adjustments.

### 3.2. Adjuncts to Enhance the Effectiveness of Anti-Myeloma Therapy

The γ-secretase enzyme cleaves a cell-surface receptor in a signaling pathway that is dysregulated in some cancers. γ-Secretase inhibition was considered a possible new approach to cancer treatment based on preclinical findings, but clinical research has shown limited utility [90]. However, the γ-secretase inhibitor nirogacestat was used in combination with the BCMA-directed bispecific antibody teclistimab in the MajesTEC trial and appeared to potentiate its effectiveness [91]. It is therefore possible that γ-secretase inhibition might prove to be a useful adjunct to other BCMA-directed modalities in the treatment of MM.

Transforming growth factor (TGF)-β, through its signaling pathway, normally acts to stop proliferation or induce apoptosis. However, this pathway is mutated in various cancers, and produces opposite effects in late stages, promoting cellular proliferation and blocking apoptosis. TGF-β pathway inhibition may therefore enhance the effectiveness of cytotoxic and immune therapies. Vactosertib (TEW-7197) is an orally available TGF-β pathway inhibitor. In a Phase Ib trial (NCT03143985), vactosertib was given in combination with pomalidomide in RRMM patients. Historically, the 6-month PFS rate with pomalidomide alone in RRMM patients has been approximately 20%; 6-month PFS with combined pomalidomide and vactosertib in this study was 80%. The combined regimen was generally well tolerated [92].

### 3.3. Halting CAR-T Cell Toxicity

CRS and other immune-mediated adverse effects commonly occur during treatment with CAR-T cells. Most cases are grade ≤3 and can be managed conservatively. In serious cases of CRS, the anti-cytokine tocilizumab, or a short course of corticosteroids, may be required. A novel potential strategy for halting toxicity from CAR-T therapy is the incorporation of a “suicide gene” into the CAR-T cell. At a chosen time, when adverse effects are offsetting clinical responses, administration of the dimerizing drug rimiducid (AP1903) will trigger the suicide gene to induce apoptosis in the CAR-T cells. In a preclinical study, a suicide gene was incorporated into anti-SLAMF7 CAR-T cells; the CAR-T cells eliminated SLAMF7-positive tumor cells in mice and were, themselves, eliminated when rimiducid was administered [93].

### 3.4. Targeting a Genetic Driver of MM

The t(4;14) translocation is a high-risk cytogenetic feature found in approximately 20% of MM patients. All patients with this translocation show overexpression of the multiple myeloma SET domain protein (MMSET, also known as NSD2/WHSC1), which causes the accelerated malignant transformation of plasma cells [94]. KTX-1001, a first-in-class oral MMSET inhibitor, performed well in preclinical research and is now being assessed in an international Phase I trial in RRMM patients (NCT05651932). Dose escalation will be assessed in RRMM patients; dose expansion will then be assessed in RRMM patients exhibiting the t(4;14) translocation [95]. 

### 3.5. Beyond Bispecific

The novel tri-specific T cell activating construct HPN217 (TriTAC^®^) binds to BCMA on MM cells and CD3 on T cells, and also binds to serum albumin, which extends its half-life and could therefore result in longer DOR. In a Phase I dose-escalation trial in RRMM patients (NCT04184050), the most common treatment-emergent adverse events across all dosages in 94 treated patients were anemia, fatigue, and neutropenia. At the highest dosages, the ORR was 55%, with the majority of responses rated as very good or better [96].

### 3.6. Overcoming Anti-Apoptotic Protection in MM Cells

Like BCL-2, the related myeloid cell leukemia (MCL)-1 protein exerts anti-apoptotic effects to protect tumor cells. The overexpression of MCL-1 in MM has been associated with drug resistance and poor prognosis. The inhibition of MCL-1 may therefore prove to be a useful therapeutic strategy in MM. Several selective MCL-1 inhibitors are currently being assessed in preclinical and Phase I clinical studies [97].

### 3.7. Delivering Interferon Directly to Myeloma Cells

Modakafusp alfa (formerly TAK-573) is a cytokine that delivers interferon-α2B directly to CD38-positive myeloma cells via anti-CD38 T cells (the interferon binds to a different epitope than that to which anti-CD38 monoclonal antibodies bind). As a result of this direct delivery to the target cells, the risk of off-target adverse events is reduced. Preclinical research confirms the activation of interferon signaling in CD38-positive myeloma cells, resulting in anti-proliferative effects and direct and indirect immune cell activation. Interim data from a Phase I/II dose escalation/expansion trial of modakafusp monotherapy in RRMM patients (NCT03215030) show an ORR of 42%, a median PFS of 5.7 months, and a median DOR of 7.4 months, with comparable response rates in the subset of patients who were refractory to previous therapy with anti-CD38 monoclonal antibodies. The most frequent grade ≥3 treatment-emergent adverse events were cytopenias [98]. Further clinical trials with modakafusp are currently in progress (NCT05556616, NCT05590377).

### 3.8. Biomarker Specific Treatments

A biomarker subgroup analysis from the CANOVA study [99] showed improved efficacy with the combination of venetoclax–dexamethasone (VenDex) compared to pomalidomide–dexamethasone (PomDex) in patients with a BCL2high or gain1q mutation. The VenDex group had consistently good clinical benefit in the BCL2high and BCL2low groups. The amp(1q) mutation group had a poor prognosis with either treatment. The subgroup analysis data suggest that further tailoring of the regimen in RRMM from a vast array of newly available options based on tumor biomarkers or genomic alterations has the potential for improved efficacy and prognosis with these treatments. 

Further, a study by Rao et al. [100] demonstrated that bone marrow endothelial cells demonstrate a high level of EGFR and HB-EGF on conversion from MGUS to multiple myeloma. This study went on to show how the EGFR inhibitor erlotinib demonstrated anti-tumor activity by inhibiting receptor-mediated angiogenesis in multiple myeloma in xenograft mouse models. This could serve as a future target for targeted therapy in patients who are found to have a high expression of the pro-angiogenesis markers related to their multiple myeloma.

A study by Lu et al. [101] showed a high expressivity of the protein Aurora Kinase A (AURKA) in patients with high-risk multiple myeloma. MLN8237, a small molecule AURKA inhibitor, inhibited multiple myeloma proliferation by inducing cell injury and apoptosis. Thus, this provides another valuable biomarker-related targeted therapeutic option for the future.

### 3.9. Therapeutic Options in High-Risk Patients

The combination of proteasome inhibitors, immunomodulatory drugs, and anti-CD38 molecules have improved outcomes in RRMM; however, patients with high-risk cytogenetics and extramedullary disease still have a poor prognosis. Extramedullary disease is also associated with early progression with CAR-T cell therapy. Research is underway to improve treatment outcomes in these patients. 

The RedirecTT-1 trial [85] used the combination of GPRC5D-directed bispecific antibody Talquetamab and the BCMA-directed antibody Teclistamab in RRMM. A total of 33% of patients had high-risk genetics, 78% were triple-class refractory, and 43% had extramedullary disease. The overall response rate was 84% for the entire cohort and 73% for patients with extramedullary disease. A total of 31% of patients with extramedullary disease achieved a CR or better.

A study by Dima et al. [102] assessed outcomes with thecommercially available BCMA-directed CAR-T cell molecules Ide-cel and Cilta-cel in heavily pretreated patients with extra medullary disease (EMD). It reported an overall response rate (ORR) of 78% and a response of CR or better for 42% of the EMD group. Notable toxicities were CRS in 73%, and immune effector cell-associated neurotoxicity syndrome (ICANS) in 28%, of patients with EMD. The study did demonstrate poorer outcomes for patients with EMD compared to those without EMD, highlighting EMD as a risk factor in RRMM.

## 4. Conclusions

The fast-growing treatment landscape in relapsed refractory myeloma demonstrates significant progress marked by a wide array of novel therapies with targeted mechanisms of action or that are geared towards overcoming pathways of resistance. These novel treatment modalities range from phase I to being available for use. Importantly, several options have shown fair efficacy when used in fourth line treatment or beyond, which adds layers to the options available for patients with refractory disease in whom prognosis had previously been very dismal. 

It is hoped that future studies will be aimed towards biomarkers, genetic mutations, or patient subgroups to allow for the tailoring of treatment regimens in an optimal fashion, on an individual basis. While multiple myeloma may be a condition without a known definitive cure, the growing expanse of novel therapies in the relapsed refractory setting provides hope of deeper lines of treatment that demonstrate efficacy and tolerability. 

### Limitations

Since this is a narrative review of novel approved and experimental therapies for RRMM, this review is unable to take into account the heterogeneity of patient cohorts in the reported studies or biases involved in the studies and is unable to report on genetic or molecular profiling data besides that reported in the study itself.

## Figures and Tables

**Table 1 cancers-16-02931-t001:** Currently approved targeted therapy for relapsed/refractory multiple myeloma (RRMM).

Category	Mechanism of Action	Drugs	Indications in Multiple Myeloma and RRMM	Notes
Immuno-modulatory drugs	Marks Ikaros and Aiolos transcription factors for degradation	Lenalidomide (Revlimid^®^)	With dex, as treatment for adult pts and as maintenance following HSCT	Oral; thalidomide derivatives, contra-indicated in pregnancy
		Pomalidomide (Pomalyst^®^)	With dex, in pts who have had ≥2 prior therapies including lenalidomide and a PI, with disease progression ≤60 days of completing last therapy	
Proteasome inhibitors	Prevents breakdown of pro-apoptotic proteins	Bortezomib (Velcade^®^)	Adult pts with multiple myeloma	IV or SC
		Carfilzomib (Kyprolis^®^)	Adult RRMM pts who have had ≥1 prior therapy	IV
		Ixazomib (Ninlaro^®^)	With lenalidomide + dex in pts who have had ≥1 prior therapy	Oral
Monoclonal antibodies	Anti-CD38	Daratumumab (Darzalex^®^)	With lenalidomide or carfilzomib + dex in RRMM pts who have had 1–3 prior therapies; with pomalidomide + dex in pts who have had ≥2 prior therapies including lenalidomide and PI; as monotherapy in pts who have had ≥3 prior therapies including PI and IMiD or are double-refractory to PI and IMiD	IV, SQ
		Isatuximab (Sarclisa^®^)	With pomalidomide + dex in adult pts who have had ≥2 prior therapies including lenalidomide and PI; with carfilzomib + dex in adult RRMM pts who have had 1–3 prior therapies	IV
	Anti-SLAMF7	Elotuzumab (Empliciti^®^)	With lenalidomide + dex in adult pts who have had 1–3 prior therapies; with pomalidomide + dex in adult pts who have had ≥2 prior therapies including lenalidomide and PI	IV
XPO1 inhibitor	Forces nuclear retention of TSP	Selinexor (Xpovio^®^)	With bortezomib + dex in adult pts who have had ≥1 prior therapy; with dex in adult RRMM pts who have had ≥4 prior therapies, refractory to ≥2 PIs, ≥2 IMiDs, and anti-CD38 Mab	Oral
Bispecific antibodies (CD3 T cell engagers)	BCMA-directed	Teclistamab (Tecvayli^®^)	Adult RRMM pts who have had ≥4 prior therapies including PI, IMiD, and anti-CD38 Mab	SC; boxed warning: CRS, neurotoxicity
		Elranatamab (Elrexfio™)		
	GPRC5D-directed	Talquetamab (Talvey™)		
CAR-T cells	BCMA-directed	Ide-cel (Abecma^®^)	Adult RRMM pts who have had ≥2 prior therapies including IMiD, PI, and anti-CD38 Mab	IV; boxed warning: CRS, HLH/MAS, neurotoxicity
		Cilta-cel (Carvykti^®^)	Adult RRMM pts who have had ≥1 prior therapy including IMiD, PI, and anti-CD38 Mab	IV; boxed warning: CRS, Parkinsonism, GBS, HLH/MAS, neurotoxicity

BCMA = B-cell maturation antigen; CAR = chimeric antigen receptor; Cilta-cel = ciltacabtagene autoleucel; CRS = cytokine release syndrome; dex = dexamethasone; GBS = Guillain-Barre syndrome; HLH/MAS = hemophagocytic lymphohistiocytosis/macrophage activation syndrome; HSCT = hematopoietic stem cell transplant; Ide-cel = idecabtagene vicleucel; IMiD = immunomodulatory drug; IV = intravenous; Mab = monoclonal antibody; PI = proteasome inhibitor; pts = patients; SC = subcutaneous; TSP = tumor suppression proteins.

**Table 2 cancers-16-02931-t002:** Pivotal clinical trials for commonly used regimens in the treatment of RRMM.

Study Name	Regimen	Phase	PFS	Reference(s)
**IMiD and/or PI based doublet or triplet regimens**				
ASPIRE	KRd vs. Rd	III	mPFS 26.3 vs. 17.6 mo (*p* < 0.0001)	[1]
Tourmaline-MM1	IRd vs. Rd	III	mPFS 20.6 vs. 14.7 mo (*p* = 0.01)	[2,3]
ENDEAVOR	Kd vs. Vd	III	mPFS 18.7 vs. 9.4 mo (*p* < 0.0001)	[4]
OPTIMISMM	PVd vs. Vd	III	mPFS 11.2 vs. 7.1 mo (*p* < 0.0001)	[5]
EMN011	KPd	II	mPFS 26 mo	[6]
GEM-KyCyDex	KCd vs. Kd	II	mPFS 19.1 vs. 16.6 mo (*p* = 0.577) In Len refractory, PFS 18.4 vs. 11.3 mo (*p* = 0.043)	[7]
**Daratumumab (Dara) regimens**				
CASTOR	Dara-Vd vs. Vd	III	mPFS 16.7 vs. 7.1 mo (*p* < 0.0001)	[8]
POLLUX	Dara-Rd vs. Rd	III	mPFS 44.5 vs. 17.5 mo (*p* < 0.0001)	[9]
CANDOR	Dara-Kd vs. Kd	III	mPFS 28.6 vs. 15.2 mo (*p* < 0.0001)	[10]
APOLLO	Dara-Pd vs. Pd	III	mPFS 12.4 vs. 6.9 mo (*p* = 0.0018)	[11]
**Isatuximab (Isa) regimens**				
ICARIA	Isa-Pd vs. Pd	III	mPFS 11.5 vs. 6.5 mo (*p* = 0.001)	[12]
IKEMA	Isa-Kd vs. Kd	III	mPFS 35.7 vs. 19.2 mo	[13,14]
**Elotuzumab (Elo) regimens**				
ELOQUENT-2	Elo-Rd vs. Rd	III	mPFS 19.4 vs. 14.9 mo (*p* < 0.001)	[15]
ELOQUENT-3	Elo-Pd vs. Pd	II	mPFS 10.3 vs. 4.7 mo (*p* = 0.008)	[16]
**Bispecific antibody regimens**				
MajesTEC-1	Teclistamab in ≥4 prior lines	I/II	mPFS 11.4 mo	[17,18]
MagnetisMM-3	Elranatamab in ≥4 prior lines	II	mPFS 11.8 mo	[19,20]
MonumenTAL-1	Talquetamab in ≥4 prior lines	II	mPFS 7.5, 11.2 and 7.7 in qw, q2w, prior TCR groups	[21]
**CAR-T cell regimens**				
KarMMa-3	Ide-cel vs. SOC	III	mPFS 13.3 vs. 4.4 mo (*p* < 0.001)	[22,23]
CARTITUDE-1	Cilta-cel in ≥4 prior lines	I/II	mPFS 34.9 mo	[24,25]
CARTITUDE-4	Cilta-cel vs. SOC in ≥1 prior line Len refractory	III	mPFS NE vs. 12 mo (HR = 0.26, *p* < 0.0001) 12-mo PFS 76 vs. 49%	[26]
**Other regimens**				
BOSTON	XVd vs. Vd	III	mPFS 13.9 vs. 9.5 mo (*p* = 0.0075)	[27,28]
BELLINI	Ven-Vd vs. Vd	III	mPFS 23.4 vs. 11.4 mo In t(11;14), mPFS 36.8 vs. 9.3 mo	[29]
DREAMM-8	BPd vs. PVd	III	12-mo PFS 71 vs. 51%	[30]

IMiD: immunomodulatory drugs, PI: proteasome inhibitor, KRd: carfilzomib + lenalidomide + dexamethasone, Rd: lenalidomide + dexamethasone, mPFS: median progression-free survival, mo: months, IRd: ixazomib + lenalidomide + dexamethasone, Kd: carfilzomib + dexamethasone, Vd: Bortezomib + dexamethasone, PVd: pomalidomide + bortezomib + dexamethasone, KPd: pomalidomide + carfilzomib + dexamethasone, KCd: carfilzomib + cyclophosphamide + dexamethasone, Len: lenalidomide, Dara-Vd: daratumumab + bortezomib + dexamethasone, Dara-Rd: daratumumab + lenalidomide + dexamethasone, Dara-Kd: daratumumab + carfilzomib + dexamethasone, Dara-Pd: daratumumab + pomalidomide + dexamethasone, Pd: pomalidomide + dexamethasone, Isa-Pd: isatuximab + pomalidomide + dexamethasone, Isa-Kd: Isatuximab + carfilzomib + dexamethasone, Elo-Rd: elotuzumab + lenalidomide + dexamethasone, Elo-Pd: elotuzumab + pomalidomide + dexamethasone, qw: every week, q2w: every 2 weeks, TCR: t cell receptor therapy, CAR-T cell: chimeric antigen receptor T cell, Ide-cel: idecabtagene vicleucel, SOC: standard of care, Cilta-cel: ciltacabtagene autoleucel, NE: not evaluable, XVd: selinexor + bortezomib + dexamethasone, Ven-Vd: venetoclax + bortezomib + dexamethasone, t(11;14): translocation 11;14, BPd: belantamab mafodotin + pomalidomide + dexamethasone.

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
