# Peer review of "Updates on Therapeutic Strategies in the Treatment of Relapsed/Refractory Multiple Myeloma"

_cancers, 2024, doi:10.3390/cancers16172931_

Round 1

Reviewer 1 Report

Comments and Suggestions for Authors

The manuscript provides an overview of targeted therapies for relapsed/refractory multiple myeloma (RRMM). While the topic is significant and relevant, the manuscript falls short in several critical areas.

1- The manuscript primarily reviews well-known therapies and does not offer new insights or advancements in the field. The content is largely a compilation of existing information without providing a fresh perspective or new data that could benefit the scientific community. Novel studies and recent advancements are noticeably missing from the discussion.

https://doi.org/10.1200/JCO.24.01008

DOI: 10.1056/NEJMoa2403407 (DREAMM-8)

https://doi.org/10.1200/JCO.2024.42.16_suppl.7510

2- The manuscript lacks any figures that could help summarize key points and enhance the reader's understanding. Including visual aids to represent complex data and mechanisms would significantly improve the manuscript's readability and impact.

3- Please indicate why you need to perform this review and how it differs from others , although there are many new reviews on this subject. ( e.g. https://doi.org/10.3390/ijms25116192, https://doi.org/10.3390/ph16111628)

Comments on the Quality of English Language

Minor editing of English language required

Author Response

Reviewer 1

The manuscript provides an overview of targeted therapies for relapsed/refractory multiple myeloma (RRMM). While the topic is significant and relevant, the manuscript falls short in several critical areas.

1- The manuscript primarily reviews well-known therapies and does not offer new insights or advancements in the field. The content is largely a compilation of existing information without providing a fresh perspective or new data that could benefit the scientific community. Novel studies and recent advancements are noticeably missing from the discussion.

https://doi.org/10.1200/JCO.24.01008

DOI: 10.1056/NEJMoa2403407 (DREAMM-8)

https://doi.org/10.1200/JCO.2024.42.16_suppl.7510

Reviewer Reply:

We thank you for your comment.  The publications above were published after the date of our initial submission. We are grateful to be able to add these to our review. They are included in lines 491-498 and 538-545.

2- The manuscript lacks any figures that could help summarize key points and enhance the reader's understanding. Including visual aids to represent complex data and mechanisms would significantly improve the manuscript's readability and impact.

Reviewer Reply:

We thank you for your comment. We have added a figure that would augment understanding of mechanism of action of described therapeutic modalities.

3- Please indicate why you need to perform this review and how it differs from others , although there are many new reviews on this subject. ( e.g. https://doi.org/10.3390/ijms25116192, https://doi.org/10.3390/ph16111628)

Reviewer Reply:

We thank you for your comment. We feel that compared to the reviews above and other reviews on this subject, our review includes a wider scope of treatment modalities in the relapsed refractory setting and adds important, updated information on outcome and tolerability data. This includes treatment modalities published at ASH conference 2023 as well as addition of information from ASCO 2024 to this latest version.

Reviewer 2 Report

Comments and Suggestions for Authors

The manuscript The Current Landscape in Targeted Therapy for 2 Relapsed/Refractory Multiple Myeloma summarizes the vast amount of data from recent trials of currently approved and investigational agents, classified by mechanism of action, and describes some innovative new approaches. To my knowledge, it is one of the best reviews on the topic, very well organized and written in a clear, concise language.

I have only one minor concern: the manuscript lacks a small general conclusion after the last section (3. Future directions) of the manuscript. The last section briefly outlines the perspectives. However, a short general conclusion would strengthen the MS.

Author Response

The manuscript The Current Landscape in Targeted Therapy for 2 Relapsed/Refractory Multiple Myeloma summarizes the vast amount of data from recent trials of currently approved and investigational agents, classified by mechanism of action, and describes some innovative new approaches. To my knowledge, it is one of the best reviews on the topic, very well organized and written in a clear, concise language.

I have only one minor concern: the manuscript lacks a small general conclusion after the last section (3. Future directions) of the manuscript. The last section briefly outlines the perspectives. However, a short general conclusion would strengthen the MS.

Reviewer Reply:

We thank you for your kind comment and support. We have taken your concern into consideration and added a general conclusion after the last section. Thank you.

Reviewer 3 Report

Comments and Suggestions for Authors

According to my best knowledge, the prognosis of multiple myeloma has improved significantly in recent years. There is no doubt that definitive curative treatment is not available for most patients, but the content and tone of this publication are so negative for me that I would not recommend its publication.

Author Response

According to my best knowledge, the prognosis of multiple myeloma has improved significantly in recent years. There is no doubt that definitive curative treatment is not available for most patients, but the content and tone of this publication are so negative for me that I would not recommend its publication.

Reviewer Reply:

We thank you for your comment. We reflected on our manuscript and while it was not our intention, we agree that the tone of our manuscript can come off as negative and have thus adjusted the tone where appropriate as these novel therapies are positive developments in the treatment of relapsed refractory multiple myeloma.

Reviewer 4 Report

Comments and Suggestions for Authors

The authors discuss the complex and challenging landscape of Multiple Myeloma, a disease marked by the abnormal growth of monoclonal B-cells, for which no cure exists. Patients often experience relapses following initial treatment, and the prognosis is grim, especially for those who do not respond to first-line therapies. The chances of achieving remission decrease with each subsequent relapse. However, the authors note a glimmer of hope in the expanding array of targeted treatments, which may offer responses to patients who have not benefited from older therapies. These new treatments are frequently used in combination with other modalities. The document provides an overview of the currently approved and experimental drugs, categorized by their mechanisms of action, and highlights innovative strategies being studied in both laboratory and clinical contexts.

Here are the key limitations identified in the study and some suggestions:

Unadjusted confounding by clinical factors: Some heterogeneity between cohorts could not be addressed, as patient characteristics like disease stage/risk were not always reported uniformly. Better adjustment for confounders would strengthen conclusions.

Immune cell assessment methods varied: Studies used IHC, flow cytometry and gene expression, which could affect comparability. Standardizing detection techniques could improve consistency.

Expression data from plasma cells alone was analyzed rather than full microenvironment: this may not fully capture immune interactions in the bone marrow niche. More comprehensive tissue profiling is needed.

Relative lack of data on immune cell functional states: Phenotypic characterization of subsets like M1/M2 macrophages and polarized neutrophils/mast cells was largely missing. Incorporating functional markers could provide deeper insights.

Publication bias cannot be ruled out: Though statistical tests were not significant, asymmetries in funnel plots suggest some bias. Stricter protocols for identification and inclusion of studies may help address this.

Mechanistic insights are limited: Clinical outcome correlations do not prove causality. Further exploration of immune-myeloma bidirectional signaling loops through preclinical models could strengthen understanding.

Here are some suggestions for improving a future version of this systematic review and meta-analysis:

Expand search strategies to include more databases/sources without date restrictions to minimize publication bias.

Develop a detailed review protocol specifying eligibility criteria, data extraction fields, outcomes of interest, etc. to ensure reproducibility.

Collect detailed individual patient data where possible to allow for more adjusted statistical analysis accounting for clinical/demographic covariates.

Include studies assessing immune profiles in serum/plasma in addition to bone marrow to correlate systemic and microenvironment immunity.

Categorize immune cell phenotypes (e.g. M1 vs M2 macrophages) and functional states (exhausted vs stem-like T cells) using standardized classifiers.

Develop collaboration with study authors to obtain raw immune profiling data for re-analysis using harmonized computational methods (e.g. gene signatures).

Perform network-based analysis to map correlations between multiple immune subsets and myeloma phenotypes.

Incorporate outcomes beyond survival like MRD response, relapse patterns to characterize full disease course.

If not in the scope of the manuscript, highlight in limitations

This reviewer personally misses 3 aspects:

I. is there any evidence if  In high-risk patients (cytogenetic, extramedullary disease,etc.) with suboptimal response afterwards

Is therapy intensification useful?In high-risk patients with suboptimal response to initial therapies, intensification of therapy may be considered. This could involve the use of high-dose chemotherapy followed by autologous stem cell transplantation, the addition of novel agents to the treatment regimen, or the use of more aggressive combination therapies. The decision to intensify therapy should be based on the patient's overall health, the specific characteristics of their disease, and the potential benefits and risks of the treatment options.

Clinical evidence for the utility of therapy intensification in high-risk MM patients comes from various studies and clinical trials. For example, the use of proteasome inhibitors, immunomodulatory drugs, and monoclonal antibodies in combination with standard therapies has been shown to improve outcomes in high-risk patients. However, the optimal approach may vary depending on individual patient factors, and close monitoring for toxicity and response is essential.

II.again in the context of future directions, future directions, angiogenesis, the formation of new blood vessels, is crucial for the growth and spread of tumors. MM cells can induce angiogenesis to support their proliferation and survival. Anti-angiogenic therapies aim to disrupt this process by targeting vascular endothelial growth factor (VEGF) and its receptor (VEGFR), among other pathways. Bevacizumab, a monoclonal antibody against VEGF, and other anti-angiogenic agents have been explored in MM, with some success in reducing disease progression.

As a fitting example, HB-EGF (heparin-binding epidermal growth factor-like growth factor) axis into the treatment of multiple myeloma (MM) could potentially offer additional therapeutic strategies. The HB-EGF axis involves the interaction between HB-EGF and its receptor, the epidermal growth factor receptor (EGFR), which plays a role in cell proliferation, survival, and migration.

In the context of MM, dysregulation of the HB-EGF axis may contribute to the pathogenesis of the disease. Targeting this axis could involve the use of agents that inhibit HB-EGF or its receptor, thereby disrupting the signaling pathways that support MM cell growth and survival.

To integrate HB-EGF axis targeting with other therapeutic approaches sa combination strategy could be employed. For example:

  1. Combining HB-EGF axis inhibitors with Bites or conventional therapy could potentially enhance the anti-proliferative effects on MM cells by targeting both cell cycle regulation and growth factor signaling.

  2. Integrating HB-EGF axis inhibitors with anti-angiogenic therapies might synergistically reduce the tumor's ability to promote new blood vessel formation, as HB-EGF has been implicated in angiogenesis.

  3. In high-risk MM patients with suboptimal response to initial therapies, adding HB-EGF axis inhibitors to more intensive treatment regimens could provide additional benefit by attacking the cancer from another angle.

However, it is important to note that the integration of HB-EGF axis targeting into MM treatment is speculative and would require clinical evidence to support its efficacy and safety. Preclinical and clinical studies would be necessary to determine the best approaches for combining HB-EGF axis inhibitors with existing therapies, as well as to understand the potential benefits and risks in different patient populations, including those with high-risk disease (please refer to PMID: 31936715 and expand the introduction and discussion sections).

III. lastly,  individualized therapy based on patient-specific clinical characteristics, optimal management of MM complications, and the development of novel therapeutic options. Among these, targeting aurora kinase A (AURKA) and mitotic pathways, as well as angiogenesis, are potential strategies that could be integrated into the treatment paradigm.

AURKA is a serine/threonine kinase involved in the regulation of mitosis, and its overexpression has been associated with various cancers, including MM. Inhibitors of AURKA can disrupt the cell cycle and induce apoptosis in cancer cells. Preclinical studies have shown that AURKA inhibitors can be effective against MM cells, particularly in combination with other therapies. For example, the AURKA inhibitor alisertib has been investigated in MM and has shown some promise in clinical trials.

Mitotic targeting can be achieved through various mechanisms, including the inhibition of microtubule dynamics, which is a well-established strategy in cancer therapy. Drugs like taxanes and vinca alkaloids, which interfere with microtubule function, have been used in MM treatment. Novel agents that target mitotic pathways are also being developed and may offer new avenues for therapy.

Comments on the Quality of English Language

Good

Author Response

Reviewer 4

The authors discuss the complex and challenging landscape of Multiple Myeloma, a disease marked by the abnormal growth of monoclonal B-cells, for which no cure exists. Patients often experience relapses following initial treatment, and the prognosis is grim, especially for those who do not respond to first-line therapies. The chances of achieving remission decrease with each subsequent relapse. However, the authors note a glimmer of hope in the expanding array of targeted treatments, which may offer responses to patients who have not benefited from older therapies. These new treatments are frequently used in combination with other modalities. The document provides an overview of the currently approved and experimental drugs, categorized by their mechanisms of action, and highlights innovative strategies being studied in both laboratory and clinical contexts.

Here are the key limitations identified in the study and some suggestions:

Unadjusted confounding by clinical factors: Some heterogeneity between cohorts could not be addressed, as patient characteristics like disease stage/risk were not always reported uniformly. Better adjustment for confounders would strengthen conclusions.

 Reviewer reply: We thank you for your comment. We do agree with your point. However, this is a narrative review and non-uniformity of the cohorts in the various studies is not in the scope of this review to address. Our goal is to summarize novel therapeutic options, approved and experimental, for relapsed refractory multiple myeloma.

Immune cell assessment methods varied: Studies used IHC, flow cytometry and gene expression, which could affect comparability. Standardizing detection techniques could improve consistency.

 Reviewer reply: We thank you for your comment. We do agree with your point. However, this is a narrative review and the molecular detection techniques used is not in the scope of this review to address. Our goal is to summarize novel therapeutic options, approved and experimental, for relapsed refractory multiple myeloma.

 Expression data from plasma cells alone was analyzed rather than full microenvironment: this may not fully capture immune interactions in the bone marrow niche. More comprehensive tissue profiling is needed.

 Reviewer reply: We thank you for your comment. We do agree with your point. However, this is a narrative review and the tissue profiling used is not in the scope of this review to address. Our goal is to summarize novel therapeutic options, approved and experimental, for relapsed refractory multiple myeloma.

Relative lack of data on immune cell functional states: Phenotypic characterization of subsets like M1/M2 macrophages and polarized neutrophils/mast cells was largely missing. Incorporating functional markers could provide deeper insights.

Reviewer reply: We thank you for your comment. We do agree with your point. However, this is a narrative review and we would be unable to incorporate functional immune cell markers in our study. Our goal is to summarize novel therapeutic options, approved and experimental, for relapsed refractory multiple myeloma.

Publication bias cannot be ruled out: Though statistical tests were not significant, asymmetries in funnel plots suggest some bias. Stricter protocols for identification and inclusion of studies may help address this.

Mechanistic insights are limited: Clinical outcome correlations do not prove causality. Further exploration of immune-myeloma bidirectional signaling loops through preclinical models could strengthen understanding.

Reviewer reply: We thank you for your comment. We do agree with your point. We will mention potential bias in the study population for reported studies as a limitation in our review. Exploration of the immune-myeloma bidirectional signaling loop through preclinical models is beyond the scope of our review.

Here are some suggestions for improving a future version of this systematic review and meta-analysis:

Expand search strategies to include more databases/sources without date restrictions to minimize publication bias.

Develop a detailed review protocol specifying eligibility criteria, data extraction fields, outcomes of interest, etc. to ensure reproducibility.

Collect detailed individual patient data where possible to allow for more adjusted statistical analysis accounting for clinical/demographic covariates.

Include studies assessing immune profiles in serum/plasma in addition to bone marrow to correlate systemic and microenvironment immunity.

Categorize immune cell phenotypes (e.g. M1 vs M2 macrophages) and functional states (exhausted vs stem-like T cells) using standardized classifiers.

Develop collaboration with study authors to obtain raw immune profiling data for re-analysis using harmonized computational methods (e.g. gene signatures).

Perform network-based analysis to map correlations between multiple immune subsets and myeloma phenotypes.

Incorporate outcomes beyond survival like MRD response, relapse patterns to characterize full disease course.

If not in the scope of the manuscript, highlight in limitations

Reviewer reply: We thank you for your comment. We do agree with your point. However, this is a narrative review. Our goal is to summarize novel therapeutic options, approved and experimental, for relapsed refractory multiple myeloma. We have not performed statistical tests on pooled studies and are unable to incorporate information about raw immune profiling, immune cell phenotypes, functional states etc. We will mention this as a limitation in our study.

This reviewer personally misses 3 aspects:

  1. is there any evidence if In high-risk patients (cytogenetic, extramedullary disease,etc.) with suboptimal response afterwards

Reviewer reply: We thank you for your comment. The data used for high-risk patients was preliminary or early in its use hence it does not, as yet, capture robust data on response at this time.

Is therapy intensification useful?In high-risk patients with suboptimal response to initial therapies, intensification of therapy may be considered. This could involve the use of high-dose chemotherapy followed by autologous stem cell transplantation, the addition of novel agents to the treatment regimen, or the use of more aggressive combination therapies. The decision to intensify therapy should be based on the patient's overall health, the specific characteristics of their disease, and the potential benefits and risks of the treatment options.

Reviewer reply: We thank you for your comment. We incorporated data on therapies for high-risk RRMM patients under a new subheading including the RedirecTT-1 trial and data for commercially available CAR-T cell use in this population, lines 586 to 606 of the manuscript.

Clinical evidence for the utility of therapy intensification in high-risk MM patients comes from various studies and clinical trials. For example, the use of proteasome inhibitors, immunomodulatory drugs, and monoclonal antibodies in combination with standard therapies has been shown to improve outcomes in high-risk patients. However, the optimal approach may vary depending on individual patient factors, and close monitoring for toxicity and response is essential.

II.again in the context of future directions, future directions, angiogenesis, the formation of new blood vessels, is crucial for the growth and spread of tumors. MM cells can induce angiogenesis to support their proliferation and survival. Anti-angiogenic therapies aim to disrupt this process by targeting vascular endothelial growth factor (VEGF) and its receptor (VEGFR), among other pathways. Bevacizumab, a monoclonal antibody against VEGF, and other anti-angiogenic agents have been explored in MM, with some success in reducing disease progression.

As a fitting example, HB-EGF (heparin-binding epidermal growth factor-like growth factor) axis into the treatment of multiple myeloma (MM) could potentially offer additional therapeutic strategies. The HB-EGF axis involves the interaction between HB-EGF and its receptor, the epidermal growth factor receptor (EGFR), which plays a role in cell proliferation, survival, and migration.

In the context of MM, dysregulation of the HB-EGF axis may contribute to the pathogenesis of the disease. Targeting this axis could involve the use of agents that inhibit HB-EGF or its receptor, thereby disrupting the signaling pathways that support MM cell growth and survival.

To integrate HB-EGF axis targeting with other therapeutic approaches sa combination strategy could be employed. For example:

  1. Combining HB-EGF axis inhibitors with Bites or conventional therapy could potentially enhance the anti-proliferative effects on MM cells by targeting both cell cycle regulation and growth factor signaling.
  2. Integrating HB-EGF axis inhibitors with anti-angiogenic therapies might synergistically reduce the tumor's ability to promote new blood vessel formation, as HB-EGF has been implicated in angiogenesis.
  3. In high-risk MM patients with suboptimal response to initial therapies, adding HB-EGF axis inhibitors to more intensive treatment regimens could provide additional benefit by attacking the cancer from another angle.

However, it is important to note that the integration of HB-EGF axis targeting into MM treatment is speculative and would require clinical evidence to support its efficacy and safety. Preclinical and clinical studies would be necessary to determine the best approaches for combining HB-EGF axis inhibitors with existing therapies, as well as to understand the potential benefits and risks in different patient populations, including those with high-risk disease (please refer to PMID: 31936715 and expand the introduction and discussion sections).

Reviewer reply: We thank you for your comment. We agree with your point and have incorporated data on the targeting of VEGF/HB-EGF into our review. Change made at lines 574-580.

III. lastly,  individualized therapy based on patient-specific clinical characteristics, optimal management of MM complications, and the development of novel therapeutic options. Among these, targeting aurora kinase A (AURKA) and mitotic pathways, as well as angiogenesis, are potential strategies that could be integrated into the treatment paradigm.

AURKA is a serine/threonine kinase involved in the regulation of mitosis, and its overexpression has been associated with various cancers, including MM. Inhibitors of AURKA can disrupt the cell cycle and induce apoptosis in cancer cells. Preclinical studies have shown that AURKA inhibitors can be effective against MM cells, particularly in combination with other therapies. For example, the AURKA inhibitor alisertib has been investigated in MM and has shown some promise in clinical trials.

Mitotic targeting can be achieved through various mechanisms, including the inhibition of microtubule dynamics, which is a well-established strategy in cancer therapy. Drugs like taxanes and vinca alkaloids, which interfere with microtubule function, have been used in MM treatment. Novel agents that target mitotic pathways are also being developed and may offer new avenues for therapy.

Reviewer reply: We thank you for your comment. We agree with your point and have incorporated data on the targeting of AURKA into our review. Change made at lines 581-58.

Round 2

Reviewer 1 Report

Comments and Suggestions for Authors

I am satisfied that the authors have addressed all of my previous concerns about the article. It is now much improved and I feel that it is now suitable for publication.

Comments on the Quality of English Language

Minor editing of English language required

Author Response

We thank you for your kind comment and support.

Reviewer 4 Report

Comments and Suggestions for Authors

The authors have clarified several of the questions I raised in my previous review. Most of the major problems have been addressed by this revision. 

Comments on the Quality of English Language

The authors have clarified several of the questions I raised in my previous review. Most of the major problems have been addressed by this revision. 

Author Response

(The authors gave the same response as above.)
